# Botulinum toxin injection without electromyographic guidance in consecutive esotropia

Hee Kyung Yang[‡], Dong Hyun Kim[‡], Jeong-Min Hwang*

Department of Ophthalmology, Seoul National University College of Medicine, Seoul National University Bundang Hospital, Seongnam, Korea

‡ These authors share first authorship on this work.
* hjm@snu.ac.kr

## Abstract

### Purpose

To investigate the efficacy of botulinum toxin injection without electromyographic guidance for the treatment of consecutive esotropia.

### Methods

A retrospective study was performed on 49 subjects with consecutive esotropia who received botulinum toxin injection in the medial rectus muscles without the use of electro-myographic guidance. Treatment was considered successful if the final ocular alignment was orthotropic or esodeviation was ≤10 prism diopters (PD) during distant fixation.

### Results

The mean age was 15.2 ± 8.3 years. The mean esodeviation before injection was 21.8 ± 9.1 PD at distance and 21.3 ± 8.3 PD at near. The mean number of injections per patient was 1.3 ± 0.7, and 46 patients (93.9%) received two or fewer injections. At 6 months after the final injection, the mean angle of esodeviation was 7.3 ± 6.0 PD at distance and 7.5 ± 6.6 PD at near (all p<0.001), and 69.4% showed successful alignment. By multivariate analysis, an initial postoperative esodeviation of ≤18 PD at one month after exotropia surgery was considered to be a predictive factor for successful botulinum toxin injection ($P = 0.007$). Vertical deviation and/or ptosis occurred in 4 patients (8.2%) at two weeks after injection, which all resolved within three months. There was no recurrence of exotropia up to the final follow-up examination.

### Conclusion

Botulinum toxin injection without electromyographic guidance is safe and effective in the treatment of consecutive esotropia without causing recurrent exotropia. Successful botulinum toxin injection is likely in patients with an initial postoperative esodeviation of 18PD or less at one month after exotropia surgery.

**Data Availability Statement:** The data underlying the results presented in the study are partially restricted by the Institutional Review Board of Seoul National University Bundang Hospital/Ethics committee to protect patient identities. However,

the data are available upon request for researchers who meet the criteria for access. Requests may be sent to the SNUBH IRB office at 82-31-787-8801 or 02119@snubh.org.

**Funding:** This work was supported by the National Research Foundation of Korea(NRF) grant funded by the Korean government, Ministry of Science and ICT(MSIT) (No. 2020R1F1A1074481). The funding organization had no role in study design, data collection and analysis, or decision to publish.

**Competing interests:** The authors have declared that no competing interests exist.

## Introduction

Initial overcorrection is a common outcome of surgical treatment of intermittent exotropia, as it is considered as a good prognostic factor for long-term surgical success of intermittent exotropia [1, 2]. While most patients with initial postoperative overcorrection gradually improve due to postoperative exodrift after exotropia surgery [3, 4], persistent consecutive esotropia may be intolerable to patients or their parents. Persistent consecutive esotropia could be managed in the following manner. First, prism glasses could be helpful to maintain fusion; however, despite its advantage, it requires considerable time to wean off of prism glasses [4–6]. Second, long-standing persistent consecutive esotropia may ultimately require additional surgery; however, long-term results of surgery for consecutive esotropia remain highly variable and unpredictable [7]. Finally, botulinum toxin injection could be an effective alternative to surgery for the management of consecutive esotropia [8–14].

Botulinum toxin injection is usually performed under electromyographic guidance, which can take up to 15 minutes [15] and may not be well tolerated in young children. In contrast, botulinum toxin injection without electromyographic guidance is practically easy to perform in the office only with topical anesthesia which offers greater simplicity, cost efficiency, and time efficiency, and it has been proven to be effective in infantile esotropia [16] and abducens nerve palsy [17]. However, to the best of our knowledge, the effect of botulinum toxin injection without electromyographic guidance in consecutive esotropia has not been demonstrated in a large cohort study. Therefore, the aim of this study was to evaluate the effect of botulinum toxin injection in patients with consecutive esotropia without the use of electromyography.

## Materials and methods

We reviewed the medical records of patients with consecutive esotropia that occurred after exotropia surgery, who had undergone botulinum toxin A (Innotox®, Medytox Inc., Korea) injection in the medial rectus muscles under topical anesthesia without electromyographic guidance during the period from April 2014 to May 2019. Patients who had paralytic or restrictive strabismus, other ocular disease that could affect vision, and systemic or neurologic diseases that could affect ocular alignment were excluded. Written consent was obtained from the patient or parent and/or legal guardian for botulinum toxin injection. This study was conducted in compliance with the Declarations of Helsinki and was approved by the Institutional Review Board of Seoul National University Bundang Hospital (IRB number: B-1905/541-101).

Data collected from the patients' medical records were age, sex, age at onset of exotropia, type of exotropia before surgery, age at previous exotropia surgery, preoperative angle of exodeviation before the last exotropia surgery, type of surgery for exotropia, immediate angle of esodeviation at one month after the last exotropia surgery, baseline angle of esodeviation before injection, age at first treatment of botulinum toxin injection, spherical equivalent refractive errors, stereoacuity, interval between exotropia surgery to initial botulinum toxin injection, initial post-treatment alignment, time between last botulinum toxin injection to last follow-up examination, number of injections and dosage of botulinum toxin, and complications after treatment. Patient records and information were anonymized and de-identified prior to the analysis.

Prism and alternate cover test at 6 m and 33 cm, respectively, in the primary position were used to measure the deviation before and after the injections. The sensory status was evaluated using the Randot stereoacuity test (Stereo Optical Company, Inc., Chicago, IL) in cooperative patients. Good stereoacuity was defined as 100 arcsec or better. Post-treatment measurements were performed at two weeks, three months, and six months after the injection. Reinjection of botulinum toxin was considered at 6 months after the initial injection if the angle of

esodeviation was > 10 PD during distant fixation. Patients with a follow-up duration of at least 6 months after their last botulinum toxin injection were included for analysis.

Under topical anesthesia with proparacaine hydrochloride 0.5% (Paracaine®, Hanmi Pharmaceutical co., ltd. Seoul), botulinum toxin, without electromyographic guidance, was injected into both medial rectus muscles transconjunctivally. The type of botulinum toxin was a liquid premixed injectable toxin without the need for reconstitution. After eyelid speculum placement, patients were instructed to look to the temporal side to expose the nasal conjunctiva. We grasped the nasal conjunctiva with forceps, and the needle was aimed inferonasally at a tangent to the globe with the bevel facing upward. Botulinum toxin was slowly injected for 20–30 seconds using a 30-gauge needle into the nasal conjunctiva at approximately 8–10 mm from the limbus closely targeted at the belly of the medial rectus muscle. The same procedure was performed in both medial rectus muscles. The total dosage of botulinum toxin injection in both medial rectus muscles per patient ranged from 6–12 IU, which was determined empirically based on their esodeviation, which ranged from 10 to 45 prism diopters (PD). Patients were instructed not to rub their eyes and to remain upright for at least one hour.

Treatment was considered successful if the ocular alignment was orthotropic or esodeviation was ≤ 10 PD during distant fixation. Patients with an esodeviation of > 10 PD during distant fixation were classified as the failure group. Complications, including ptosis, vertical deviation and overcorrection, which was defined as an exodeviation of ≥ 10 PD, were recorded at each visit.

All data were analyzed using SPSS software version 21.0 (SPSS Inc., Chicago, IL, USA). Paired t-test was used, comparing the significant differences between angles of esodeviation before and after injection. Student's t-test, $\chi$2 test, and Fisher's exact test were used to compare the patients' characteristics and treatment outcomes. Multivariate logistic regression was performed to identify the factors affecting the success after botulinum toxin injection, including preoperative angle of exodeviation at distance before the previous surgery, age at onset of exotropia, type of exotropia before surgery, age at previous exotropia surgery, spherical equivalent refractive errors, stereoacuity, interval between surgery and injection, age at first treatment of botulinum toxin injection, immediate angle of esodeviation at one month after the last exotropia surgery, baseline angle of esodeviation before injection, initial post-treatment alignment, dosage, and number of injections. A p-value of less than 0.05 was considered statistically significant. Data are presented as the means ± standard deviation, unless stated otherwise.

## Results

The clinical characteristics of consecutive esotropia patients who underwent botulinum toxin injection are presented in Table 1. A total of 49 patients with consecutive esotropia had undergone botulinum toxin injection without the use of electromyographic guidance in both medial rectus muscles. Among them, 22 patients (44.9%) were male and 27 (55.1%) were female. The mean age of patients was 15.2 ± 8.3 years (range, 7–66 years). The mean interval from strabismus surgery to the time of botulinum toxin injection was 49.8 ± 37.2 months (range, 4–150 months). The number of previous exotropia surgeries was one in 26 patients (53.1%), two in 20 patients (40.8%), and three or more in 3 patients (6.1%). Two patients (4.1%) had underwent unilateral medial rectus muscle recession for consecutive esotropia, which did not result in adequately reduced deviation.

The mean number of botulinum toxin injection per patient was 1.3 ± 0.7 (1–4), and the total dosage per patient was 8.5 ± 2.5 IU (6.0–12.0). A total of 46 patients (93.9%) received two or fewer injections; of them, 37 patients (75.5%) received one injection, 9 patients (18.4%) two injections, one patient (2.0%) three injections, and two patients (4.1%) four injections.

**Table 1. Demographics and clinical characteristics of consecutive esotropia patients who underwent botulinum toxin injection without electromyographic guidance.**

| Variable | | |
|---|---|---|
| Number of patients | | 49 |
| Mean age at treatment (y) | | 15.2 ± 8.3 |
| Sex (Male: Female) | | 22: 27 |
| Distant exodeviation before last operation (PD) | | 18.4 ± 6.4 |
| Near exodeviation before last operation (PD) | | 19.7 ± 8.5 |
| Type of exotropia before surgery | | |
| | Basic | 38 |
| | Divergence excess | 3 |
| | Convergence insufficiency | 8 |
| Type of first exotropia surgery | | |
| | Unilateral LR recession | 5 |
| | Bilateral LR recession | 12 |
| | Unilateral R&R | 32 |
| Type of recurrent exotropia surgery | | |
| | Unilateral R&R | 15 |
| | MR resection | 11 |
| Mean number of exotropia surgery | | 1.5 ± 0.7 |
| From last surgery to first injection (m) | | 49.8 ± 37.2 |
| From last injection to last follow-up (m) | | 17.6 ± 10.1 |
| Distant esodeviation before first injection (PD) | | 21.8 ± 9.1 |
| Near esodeviation before first injection (PD) | | 21.3 ± 8.3 |
| Mean number of injections | | 1.3 ± 0.7 |
| Mean dose of botulinum toxin (IU) | | 8.5 ± 2.5 |

y = years; PD = prism diopters; m = months; LR = lateral rectus; MR = medial rectus; R&R = lateral rectus recession and medial rectus resection

The mean amount of esodeviation before injection was 21.8 ± 9.1 PD at distance and 21.3 ± 8.3 PD at near. At 6 months after the final injection, the mean esodeviation was 7.3 ± 6.0 PD at distance and 7.5 ± 6.6 PD at near (all $p < 0.001$, paired t-test), and 69.4% showed successful alignment at 6 months after the final injection.

At the final follow-up examination of 17.6 ± 10.1 months (range, 6–41 months) after the last botulinum toxin injection, the mean angle of esodeviation was 7.3 ± 6.0 PD at distance and 6.9 ± 7.2 PD at near (all $p < 0.001$, paired t-test). Of all patients, 36 patients (73.5%) showed successful alignment at the final examination. The mean rate of change in esodeviation from 6 months after the injection to the last follow-up examination was -0.1 ± 2.7 PD/year (range, -8~12).

Complications of vertical deviation and/or ptosis occurred in 4 patients (8.2%), and over-correction with an exotropia of $\geq 10$ PD at distance was found in 11 patients (22.4%) at two weeks after treatment. However, all complications resolved within three months. There were no other major complications or side effects, such as scleral penetration and systemic allergic reaction.

The predictive factors of success were determined by univariate comparison between the success group and the failure group at 6 months after the final botulinum injection (Table 2). The rate of patients with an initial postoperative esodeviation of $\leq 18$ PD at one month after exotropia surgery was significantly larger in the success group compared with the failure

**Table 2. Comparison of the success group and failure group at 6 months after last botulinum toxin injection.**

| | Success group (≤ 10 PD ET) (n = 34) | Failure group (> 10 PD ET) (n = 15) | P value |
|---|---|---|---|
| **Initial esodeviation ≤ 18 PD within one month after last exotropia surgery** | **24 (70.6%)** | **5 (33.3%)** | **0.014*** |
| Baseline esodeviation before injection (PD) | 21.2 ± 8.7 | 23.0 ± 10.1 | 0.540[†] |
| **Immediate success at 2 weeks after injection** | **31 (91.2%)** | **9 (60.0%)** | **0.009*** |
| **Esodeviation at 2 weeks after injection (PD)** | **-2.3 ± 11.2** | **7.8 ± 8.3** | **0.003[†]** |
| **Esodeviation at 6 months after injection (PD)** | **4.1 ± 4.1** | **14.7 ± 1.6** | **<0.001[†]** |
| Interval from last injection to last follow-up (months) | 17.8 ± 9.8 | 17.2 ± 10.7 | 0.870[†] |
| Number of operations before injection | 1.5 ± 0.7 | 1.5 ± 0.6 | 0.742[†] |
| Number of operated horizontal rectus muscles | 2.7 ± 0.9 | 2.9 ± 1.0 | 0.525[†] |
| Previous surgery of MR resection | 30 (88.2%) | 12 (80.0%) | 0.660* |
| Number of botulinum toxin injections | 1.4 ± 0.6 | 1.3 ± 0.8 | 0.611[†] |
| Dose of botulinum toxin (IU/10PD esodeviation) | 4.4 ± 2.0 | 4.5 ± 2.0 | 0.884[†] |
| Spherical equivalent refractive error (D) | -3.79 ± 3.00 | -3.97 ± 2.06 | 0.806[†] |
| Good stereoacuity (≤ 100 arcsec) | 14 (45.2%) | 5 (38.5%) | 0.682* |

Values are presented as mean ± SD unless otherwise indicated; The angles of deviation are presented in positive numbers for esotropia and negative numbers for exotropia; PD, prism diopters; ET, esodeviation; IU, international unit; D, diopters; MR, medial rectus muscle

*P value by Fisher's exact test or Pearson's Chi-square test,

[†]P value by independent t-test

group (70.6% vs 33.3%, P = 0.014). However, the baseline angle of esodeviation before injection was not significantly different between the two groups (P = 0.540). The success group, when compared with the failure group, showed a smaller angle of esodeviation at all periods up to six months (all P < 0.05, respectively). By multivariate analysis, an initial postoperative esodeviation of 18PD or less at one month after exotropia surgery (P = 0.007) was a significant factor of success at six months after botulinum toxin injection.

## Discussion

This study included the largest number of patients with consecutive esotropia treated with botulinum toxin injection without electromyographic guidance. In this study, 69.4% of consecutive esotropia patients with an average esodeviation of 21.8 ± 9.1 PD at distance achieved successful outcome after botulinum toxin injection. An initial postoperative esodeviation of 18 PD or less at one month after exotropia surgery was a significant predictor of success at six months after injection. Botulinum toxin injection without electromyographic guidance can be applied to children as young as 7 years of age. Complications, including vertical deviation and/ or ptosis, occurred in 8.2% at two weeks after injection, but all resolved within three months. The success rate of botulinum toxin injection for consecutive esotropia was comparable to surgery with a success rate of 62.5% after unilateral medial rectus recession [7]. Above all, a major advantage of botulinum toxin injection over surgery is the absence of the risk of permanent overcorrection and recurrent exotropia [7].

While botulinum toxin injection has been attempted in various types of strabismus, the highest success rate, with a rate of 93%, was observed in 14 patients with consecutive esotropia [8]. One of the reasons for such a successful outcome in patients with consecutive esotropia may be attributable to the anatomical characteristics of the medial rectus muscle. The medial rectus muscle has a larger density of synapses consisting of singly innervated fibers [18], which are profoundly affected by botulinum toxin in the acute phase [19]. Rayner et al [11] also

reported that 10 of 14 patients with consecutive esotropia benefited most from botulinum toxin injection in terms of binocularity improvement among 163 children with various causes of strabismus. Therefore, botulinum toxin injection may be a reasonable choice for the treatment of consecutive esotropia, in hopes to avoid reoperation and accelerate the speed of postoperative exodrift after exotropia surgery.

To date, predictive factors of success after botulinum toxin injection remain controversial. Among the various types of childhood strabismus, the highest success rates were seen in patients with consecutive esotropia; moreover, it was more common for deviations to be corrected to 10 PD or less in those with smaller deviations (10–20 PD) than in those with larger deviations (20–110 PD) (73% vs. 54%) [20]. Good fusion potential has been suggested to be an important factor of success after botulinum toxin injection in surgically overcorrected exotropia; 15 out of 36 patients (42%) with a good fusion potential maintained good ocular alignment and resolution of their diplopia with a single injection of botulinum toxin [12]. Meanwhile, only 4 patients (17%) with no expected fusion potential achieved good alignment [12]. In our study, the initial postoperative esodeviation of ≤18 PD at one month after exotropia surgery was considered to be the only preoperative predictive factor of success. This is an interesting finding since the baseline angle of esotropia, measured as the maximum angle of esotropia before injection, was not significant, which suggests that a structural threshold of esodeviation may be determined early after surgery and can be overcome by the natural exodrift following surgery with the help of botulinum toxin injection. Conversely, a larger angle of esodeviation exceeding the structural threshold may not be able to overcome conservatively. In addition, a good immediate response at 2 weeks after botulinum toxin injection was a significant predictor of success. In other words, patients with a poor initial response at 2 weeks are less likely to improve with time; and thus, in such patients, early strabismus surgery for consecutive esotropia should be recommended. Surgery for consecutive esotropia after botulinum toxin injection was performed in one patient in the failure group; this patient had an esotropia of 16 PD at 6 months after botulinum toxin injection. She maintained fusion with prism glasses for another 2 years. However, the esodeviation remained the same and the patient eventually underwent unilateral medial rectus recession. Other patients in the failure group were reluctant to undergo surgery and used prism glasses to maintain fusion.

The optimal time and indications for botulinum toxin injection in consecutive esotropia remain unclear [10]. Considering the natural exodrift after exotropia surgery, it is likely that small angles of esotropia may resolve spontaneously over time [3, 4]. Park et al. have shown that more than 50% of the total amount of exodrift was observed within the first postoperative year, but no subsequent changes were observed after 3 years [21]. Couser et al. reported that children treated with botulinum toxin injection within 12 months of consecutive esotropia had an excellent outcome; however, the patient with the longest delay of 53 months between the onset of consecutive esotropia and treatment also had an excellent outcome after two injections [10]. In our study, the mean interval from strabismus surgery to the time of botulinum toxin injection was 49.8 ± 37.2 months (range, 4–150 months), and the patient with the longest delay of 150 months between the time of operation and botulinum treatment also had a successful outcome after two injections. Therefore, botulinum toxin treatment should be actively considered if significant consecutive esotropia persists after 3 years. Nine patients received botulinum treatment within the first year after strabismus surgery. These patients had a relatively large angle of esodeviation, as large as up to 44PD, which is less likely to spontaneously resolve over time.

To the best of our knowledge, there are only few reports about botulinum toxin injection without electromyographic guidance. Benabent et al [22] reported a success rate of 53% in 40 patients with congenital esotropia, associated with complications of ptosis (23%), vertical

deviation (21%), and conjunctival hemorrhage (7%). There was no incidence of retrobulbar hemorrhage or scleral perforation, and they did not advocate electromyographic guidance when injecting botulinum in congenital esotropia [22]. Kao and Chao [23] reported a recovery rate of 63.6% after subtenon botulinum toxin injection in 11 patients with unilateral traumatic sixth nerve palsy, and concluded that subtenon botulinum toxin injection without electromyographic guidance may be comparable to that with electromyographic guidance. They reported no ptosis, vertical deviation, scleral perforation, or overcorrections in all cases [23]. Sanjari presented more post-injection blepharoptosis in those using electromyographic guidance [17]. Scleral perforation has only been reported in two cases after botulinum toxin injection without electromyographic guidance [24, 25]. This is also extremely rare after botulinum toxin injection with electromyographic guidance [26, 27], with an incidence of 0.002% [15]. In rabbits, Paik et al [28] found no different morphologic changes in the extraocular muscle fiber layers between direct intramuscular injection and subtenon injection of botulinum toxin. Therefore, complications after botulinum toxin injection are comparable with and without electromyographic guidance.

The benefits of botulinum toxin injection without electromyographic guidance in patients with consecutive esotropia are as follows. First, the cost of an electromyographic equipment is not necessary. Second, it takes less than 5 minutes for the injection, which may be tolerable even in young children. Third, this procedure can be performed under topical anesthesia without the need for general anesthesia. Fourth, we can use a 30 G needle instead of a 27 G needle with a coated shaft, of which the latter is used to perform botulinum toxin injections with electromyographic guidance. Finally, the chance of delayed recurrence of exotropia can be avoided, which is a major cause of long-term failure after surgical correction of consecutive esotropia with medial rectus recession [7].

There are some limitations in this study. Firstly, our study population did not include children under the age of 7 years. In addition, the decision to perform botulinum toxin injection depended on the consent of patients or parents. Therefore, our study population may not be representative of all patients with consecutive esotropia after exotropia. Second, the dosage of injection was chosen empirically, which was not consistent throughout the study. As frequent undercorrection was noted, we progressively increased the amount of botulinum toxin injection throughout the study period. Thus, the average dose of botulinum toxin injection in our study (3 to 6 IU per muscle) was slightly larger than that in previous studies using electromyographic guidance in which the mean dosage ranged between 1.25 to 5.0 IU per muscle, whereas the overall success rate in our study was comparable to the former studies ranging from 67 to 86% [8, 10]. In former studies, the dose effect of botulinum toxin injection cannot be standardized, since various types, Botox™ and Dysport™, and dosages were utilized [20]. Even in patients who had underwent previous unilateral lateral rectus recession and medial rectus resection, we injected botulinum toxin in both medial rectus muscles. This was because in the earlier phase of this study, frequent undercorrection was encountered when botulinum toxin was injected into only one medial rectus muscle even with a relatively large dose (7 IU). Therefore, we generally inject 6 IU of botulinum toxin into each medial rectus muscles for patients with an esodeviation of more than 10PD. Regarding the unequal dosage of botulinum toxin throughout the study, we incorporated the mean dosage of botulinum toxin per 10 PD of esodeviation for analysis, yet there was no significant difference in the mean dosage of botulinum toxin between the success and failure groups. Finally, the number of injections, interval, and follow-up period also varied. Reinjection of botulinum toxin was considered only in those with an esodeviation of greater than 10 PD at 6 months after the initial injection; however, some patients declined further treatment or received botulinum toxin injection at a later time. Long-term follow-up examinations (several years after the final injection) are still mandatory to

evaluate the stability of botulinum toxin injection in these patients. Meanwhile, previous studies evaluating the long-term efficacy of botulinum treatment, ranging from 2 to 5.5 years after injection, have shown equivalent results to those evaluating shorter follow-up periods ranging from 6 months to 2 years; this suggests that botulinum toxin injection may be effective for a period of 2 to 5 years in strabismic children [29]. This appears to be consistent with the results of our study, as the rate of change in the angle or esodeviation after 6 months was -0.1 ± 2.7 PD/year.

In conclusion, botulinum toxin A injection without electromyographic guidance appears to be effective in treating consecutive esotropia without the risk of recurrent exotropia. Based on our findings, patients with an initial postoperative esodeviation of 18PD or less at one month after exotropia surgery had a better chance of success at 6 months after botulinum toxin A injection.

## Author Contributions

**Conceptualization:** Jeong-Min Hwang.

**Data curation:** Hee Kyung Yang, Dong Hyun Kim.

**Formal analysis:** Hee Kyung Yang, Dong Hyun Kim.

**Investigation:** Jeong-Min Hwang.

**Methodology:** Jeong-Min Hwang.

**Resources:** Jeong-Min Hwang.

**Supervision:** Jeong-Min Hwang.

**Validation:** Jeong-Min Hwang.

**Writing – original draft:** Hee Kyung Yang, Dong Hyun Kim.

**Writing – review & editing:** Hee Kyung Yang, Jeong-Min Hwang.

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
