## [Decision Letter · Decision Letter 0]

20 Apr 2020

PONE-D-20-07281

­­­­­­­­­­­­­­­­­­­Botulinum toxin injection without electromyographic guidance in consecutive esotropia

PLOS ONE

Dear Dr. Hwang,

Thank you for submitting your manuscript to PLOS ONE. After careful consideration, we feel that it has merit but does not fully meet PLOS ONE’s publication criteria as it currently stands. Therefore, we invite you to submit a revised version of the manuscript that addresses the points raised during the review process.

We would appreciate receiving your revised manuscript by Jun 04 2020 11:59PM. To enhance the reproducibility of your results, we recommend that if applicable you deposit your laboratory protocols in protocols.io, where a protocol can be assigned its own identifier (DOI) such that it can be cited independently in the future. For instructions see: http://journals.plos.org/plosone/s/submission-guidelines#loc-laboratory-protocols

We look forward to receiving your revised manuscript.

Kind regards,

I-Jong Wang

Academic Editor

PLOS ONE

Journal Requirements:

2. In the ethics statement in the manuscript and in the online submission form, please provide additional information about the patient records used in your retrospective study. Specifically, please ensure that you have discussed whether all data were fully anonymized before you accessed them and/or whether the IRB or ethics committee waived the requirement for informed consent. If patients provided informed written consent to have data from their medical records used in research, please include this information.

b) If there are no restrictions, please upload the minimal anonymized data set necessary to replicate your study findings as either Supporting Information files or to a stable, public repository and provide us with the relevant URLs, DOIs, or accession numbers. Please see http://www.bmj.com/content/340/bmj.c181.long for guidelines on how to de-identify and prepare clinical data for publication. For a list of acceptable repositories, please see http://journals.plos.org/plosone/s/data-availability#loc-recommended-repositories

Reviewers' comments:

Reviewer's Responses to Questions

**Comments to the Author**

1. Is the manuscript technically sound, and do the data support the conclusions?

Reviewer #1: Yes

Reviewer #2: Yes

Reviewer #3: Yes

Reviewer #4: Yes

Reviewer #5: Partly

Reviewer #6: Yes

Reviewer #7: Yes

2. Has the statistical analysis been performed appropriately and rigorously? 

Reviewer #1: Yes

Reviewer #2: I Don't Know

Reviewer #3: Yes

Reviewer #4: Yes

Reviewer #5: Yes

Reviewer #6: Yes

Reviewer #7: Yes

3. Have the authors made all data underlying the findings in their manuscript fully available?

Reviewer #1: Yes

Reviewer #2: No

Reviewer #3: Yes

Reviewer #4: Yes

Reviewer #5: No

Reviewer #6: Yes

Reviewer #7: Yes

4. Is the manuscript presented in an intelligible fashion and written in standard English?

Reviewer #1: Yes

Reviewer #2: Yes

Reviewer #3: Yes

Reviewer #4: Yes

Reviewer #5: Yes

Reviewer #6: Yes

Reviewer #7: Yes

5. Review Comments to the Author

Reviewer #1: Thanks for your great sharing about using botulinum injection without EMG guide for consecutive esotropia patients. This method provides another option for the consecutive esotropia patients before they receive another surgical treatment on their extraocular muscles. However, I have several questions as followings:

1. The authors mentioned about botox injection on both medial rectus muscles. How about the cases with previous unilateral R and R (Resection and Recession) surgery? Did they also receive botox injection on their bilateral medial rectus muscles? Why not just inject on the previous operated eye only?

2. The authors didn’t mention about their dosage of botox injections in detail. How did you decide whether 6 U or 12 U for application? Is there any correlation between the dosage of botox and the success/failure effect?

3. How about the stereopsis or the suppression status of the patients? Were there any evaluations pre-op or post-op? Did better stereopsis result in better final outcome?

4. What kind of topical anesthesia agents were applied? The authors didn’t mention about it. Please clarify it.

5. How about the preoperative angle of deviation before the previous surgery, the age onset of XT, type of XT, and the age of previous XT surgery? Any clinical significance or relationships noted among these factors about success/failure?

Reviewer #2: Manuscript title: Botulinum toxin injection without electromyographic guidance in consecutive esotropia

General comments: The study aimed to investigate the efficacy of botulinum toxin injection without electromyographic assistance in treating consecutive esotropia following previous strabismus surgery. Consecutive esotropia is a condition in which strabismus is observed in the direction opposite to the initial exotropic drift prior to surgery. Many studies have reported better long-term outcomes in patients with an initial overcorrection, however overcorrection persists in some patients and reoperation may be warranted. The manuscript focuses on a relatively understudied group. They analyzed 52 subjects retrospectively and report that, with topical botulinum toxin injection without the use of electromyography, a remarkable 71.2% patients had their esotropia resolved without the need for a second surgical procedure, and only 7.7% of patient developed transient complications. The clinical effectiveness and safety appeared comparable to those with electromyographic guidance. Some issues in the methodology and the conclusion drawn from the analysis for the purpose of this study should be addressed.

1. The diagnosis of all subjects included in the study before exotropia surgery were not clearly mentioned. There are many subtypes of exotropia even after excluding paralytic and restrictive etiologies. A brief description for exotropia classification adopted in this study would help readers gain a greater understanding of study groups characteristics. Meanwhile, considering that treating muscles that had undergone previous surgery may have different effect, the type of exotropia surgery performed in the past were also not clearly introduced in the current version of manuscript. This is also important because lacking such characteristics would hinders clinical applications. A table reporting the basic demographics, subtype of exotropia, type of previous surgery, and other variables of interest may solve these issues and improve readability.

2. Injection technique without electromyographic guidance is actually a blind injection and requires practice. Although most of the complications resulted from drug extravasation were transient, the symptoms of vertical diplopia and ptosis were annoying. The authors reported a relatively low complication rates compared with previous reports. Hence the skills and tools used or avoided in the study are worth more descriptions with sufficient detail. (for instance, the methods of reconstitution of botulinum toxin, the angle of injection plane, whether or not medial rectus muscle grasping with forceps, or instructions of patients care).

3. I have two questions about the data present in the table 1. First, the PD value in the esodeviation at 6 months after injection had a much small standard deviation in the failure group ( ± 1.6 ), an explanation is needed. Second, the success group seemed to have a greater change in PD at 2 weeks from baseline compared to the failure group. The dose of botulinum toxin standardized according to the deviation angle (PD/Unit) might be different between success and failure group. A statistics analysis showing no difference between the groups is needed to exclude this possibility. Indeed a significant post-operative variables as a predictor of post-operative success may be of little clinical significance especially when the variables seemed to be highly correlated with the grouping criteria in a retrospective study design.

4. The results of multivariate logistic analysis could be represented in different methods but should not be partially revealed by a few sentences.

Reviewer #3: The authors describe the efficacy and safety of botulinum toxin injection without using EMG for consecutive ET clearly. In addition, the manuscript identifies the predictive factors of success after botulinum toxin injection. Minor points to consider in subsequent versions:

1.The manuscript highlights that immediate success at 2 weeks after botulinum injection was the only significants factor of success. Would your consider to emphasize the result in Abstract to different from other similar published literature?

2.Page 7: The predictive factors were well determined, including previous MR resection. However, the surgical methods for XT was not discuss completely in the study. In current surgical approaches, including unilateral LR recession and MR resection, bilateral LR recession and involving three or four horizontal rectus, which one might get most benefit from botulinum toxin injection? Please discuss the influence of previous surgical methods for XT.

3.Page 6: Seventy-one percents of patients received only one injection with good outcome in your study. Please analyze the risks for repeat injections.

4.Page 5: Please describe the procedure of botulinum injection more completely. Do you grape the insertion of MR muscle while injection? Some surgeon hold the needle in place for 5-10 seconds after injection. Do you do the same protocol or not? How do you adjust the distance from limbus (line 86: 8-10 mm mentioned) in children and adults (range of age: 7-66 years in Results)? Is the location of injection the same or different?

5.Page 6: The range of internal form previous strabismus surgery to the time of botulinum toxin injection was large (line 113: 4-151 months). What is your consideration in early and later intervention? Please describe in the section of Discussion.

6.Line 308: Please cite the newest reference “ Botulinum toxin for the treatment of strabismus. Cochrane Database Syst Rev. 2017 Mar 2;3:CD006499. doi: 10.1002/14651858. CD006499.pub4.” instead of the 2012 old one.

Reviewer #4: This is an interesting reserch and written properly. However, there are some questions need to be answered:

1. How was the dosage determined prior to injection? In line 88, determined by the angle of esodeviation, does it mean how many IU for how many prism diopter?

2. Did you inject unilateral or bilateral medial rectus muscles?

3. In line 87, teh 6-12 IU refers to does per injection or total injection per case? Same question goes to line 117

4. Finally, do you compare these cases to nature course? (consecutive ET nature course)

Reviewer #5: Materials and Methods

1. No information about conservative treatment (watching, patching or prism glasses) before injection. When is the optimal time of botulinum toxin injection for treating a case with consecutive esotropia?

2. Line 86 vs Line 116: dosage of botulinum toxin ranged from 6-12 IU vs 3-12 Which one is correct?

3. Line 87: the angle of esodeviation ranging from 6 to 45 PD � So you included the cases whose esodeviation angle before injection is less than 10 PD ?

Result

1. Line 110: The mean interval from strabismus surgery to the time of botulinum toxin injection was 47.3±39.o moths (range, 4-151months)

Most patients with consecutive esotropia resolves spontaneously in one year. How many cases with consecutive esotropia have botulinum toxin injection within one year (or 6 months) after strabismus surgery?

2. Please give more details of the dosage of botulinum toxin, the mean interval from strabismus surgery to the time of first injection, and the amount of esodeviation over time.

Discussion

1. You did not discuss how to exclude the possibility of spontaneous recover so that we can evaluate the efficacy of botulinum toxin properly.

2. More details about the dosage of botulinum toxin still can inspire the readers even though the dose effect cannot be standardized because of various types.

3. Overcorrection with an exotropia of ≥ 10PD at distance was found in 11 patients at two weeks after treatment but resolved within 3 months. This means nearly all overcorrected cases at 2 weeks after treatment are in failure group in the end. What do you think about this? Why dose botulinum toxin only cause permanent impact in some cases?

Reviewer #6: The concept behind the paper is interesting. That is helping ophthalmologists to choose the ideal strategy to deal with the troublesome postoperative consecutive strabismus.

1. There are lots of grammar mistakes in the article. And several repeated sentences noted (Exp. Line 67&104). Please do the grammar check.

2. The authors need to clarify in the ethics approval the IRB number.

3. The authors need to explain the rational why they performed “subconjunctival injection” rather than “subtenon injection”.

4. Is there any association between the pre-op deviation/surgical procedure and the injection outcome?

5. The short-term complication rate in this study is very low (7.7%). Ptosis/vertical deviation/overcorrection presented much frequently in literature (21-33%). Please share the pearls of reducing complications?

6. The success rate in the study is excellent. However, the 6-month-f/u time is too early to judge the long-term success. I would like to know the outcome after 12-month f/u.

Reviewer #7: The authors conducted a retrospective study on 52 subjects with consecutive esotropia who underwent botulinum toxin injection in the medial rectus muscles without the use of electromyographic guidance. At 6-months, 71.2% showed successful alignment, which is comparable to unilateral medial rectus muscle recession surgery. They also found immediate success at 2 weeks after botulinum injection is predictive of success after 6 months.

Comments

1. Did the authors investigate the type of surgery(bilateral lateral rectus muscle recession or unilateral lateral rectus muscle recession + medial rectus muscle resection/plication) before botulinum injection affect the success?

2. Why did the authors use such Botulinum doses, which is slightly larger than previous studies? Do the injection dose follow the same protocol?

6. PLOS authors have the option to publish the peer review history of their article (what does this mean?). If published, this will include your full peer review and any attached files.

Reviewer #1: No

Reviewer #2: Yes: Chia-Wei Lee

Reviewer #3: No

Reviewer #4: No

Reviewer #5: No

Reviewer #6: No

Reviewer #7: No

---

## [Author Response · Author response to Decision Letter 0]

15 Jul 2020

Reviewer #1: Thanks for your great sharing about using botulinum injection without EMG guide for consecutive esotropia patients. This method provides another option for the consecutive esotropia patients before they receive another surgical treatment on their extraocular muscles. However, I have several questions as followings:

1. The authors mentioned about botox injection on both medial rectus muscles. How about the cases with previous unilateral R and R (Resection and Recession) surgery? Did they also receive botox injection on their bilateral medial rectus muscles? Why not just inject on the previous operated eye only?

Response

Even in patients who underwent previous unilateral lateral rectus recession and medial rectus resection, we injected botulinum toxin in both medial rectus muscles. This was because in the earlier phase of this study, frequent undercorrection was encountered even with a relatively large dose (7 IU) of botulinum toxin injection into only one medial rectus muscle. Recently, we generally inject 12 IU into both medial rectus muscles for patients with an esodeviation of more than 10PD. (Lines 285-291)

2. The authors didn’t mention about their dosage of botox injections in detail. How did you decide whether 6 U or 12 U for application? Is there any correlation between the dosage of botox and the success/failure effect?

Response

We did not find a significant correlation between the amount of esotropia correction and dose of botulinum toxin injection. 

The dosage of botulinum toxin was not consistent throughout the study. As frequent undercorrection was noted, we progressively increased the amount of injection. Overall, the average dose of botulinum toxin injection in our study was slightly larger than that in previous studies using electromyographic guidance. We added the next sentence in the discussion. 

“As frequent undercorrection was noted, we progressively increased the amount of botulinum toxin injection throughout the study period.” (Lines 280-282)

3. How about the stereopsis or the suppression status of the patients? Were there any evaluations pre-op or post-op? Did better stereopsis result in better final outcome?

Response

We evaluated stereopsis in most of the patients. Sensory status was evaluated using the Randot stereoacuity test (Stereo Optical Company, Inc., Chicago, IL) in cooperative patients. Good stereoacuity was defined as 100 arcsec or better. (Lines 83-85)

However, there was no significant difference in the rate of patients with good stereopsis between the success group and failure group. (added in new Table 2)

4. What kind of topical anesthesia agents were applied? The authors didn’t mention about it. Please clarify it.

Response

Topical anesthesia was performed with proparacaine hydrochloride 0.5% (Paracaine, Hanmi Pharmaceutical co., ltd. Seoul). (Lines 90-92)

5. How about the preoperative angle of deviation before the previous surgery, the age onset of XT, type of XT, and the age of previous XT surgery? Any clinical significance or relationships noted among these factors about success/failure?

Response

Thank you for your pertinent review. We added factors including preoperative angle of exodeviation at distance before the previous surgery, age at onset of exotropia, type of exotropia before surgery and age at previous exotropia surgery. However, there was no significant difference in these factors between the success and failure groups. (Lines 113-120)

We appreciate your precious time and effort for reviewing our manuscript. 

 

Reviewer #2: Manuscript title: Botulinum toxin injection without electromyographic guidance in consecutive esotropia

General comments: The study aimed to investigate the efficacy of botulinum toxin injection without electromyographic assistance in treating consecutive esotropia following previous strabismus surgery. Consecutive esotropia is a condition in which strabismus is observed in the direction opposite to the initial exotropic drift prior to surgery. Many studies have reported better long-term outcomes in patients with an initial overcorrection, however overcorrection persists in some patients and reoperation may be warranted. The manuscript focuses on a relatively understudied group. They analyzed 52 subjects retrospectively and report that, with topical botulinum toxin injection without the use of electromyography, a remarkable 71.2% patients had their esotropia resolved without the need for a second surgical procedure, and only 7.7% of patient developed transient complications. The clinical effectiveness and safety appeared comparable to those with electromyographic guidance. Some issues in the methodology and the conclusion drawn from the analysis for the purpose of this study should be addressed.

1. The diagnosis of all subjects included in the study before exotropia surgery were not clearly mentioned. There are many subtypes of exotropia even after excluding paralytic and restrictive etiologies. A brief description for exotropia classification adopted in this study would help readers gain a greater understanding of study groups characteristics. Meanwhile, considering that treating muscles that had undergone previous surgery may have different effect, the type of exotropia surgery performed in the past were also not clearly introduced in the current version of manuscript. This is also important because lacking such characteristics would hinder clinical applications. A table reporting the basic demographics, subtype of exotropia, type of previous surgery, and other variables of interest may solve these issues and improve readability.

Response

We added a table (Table 1) of the basic demographics, subtype of exotropia, type of previous surgery, and other variables of interest.

2. Injection technique without electromyographic guidance is actually a blind injection and requires practice. Although most of the complications resulted from drug extravasation were transient, the symptoms of vertical diplopia and ptosis were annoying. The authors reported a relatively low complication rates compared with previous reports. Hence the skills and tools used or avoided in the study are worth more descriptions with sufficient detail. (for instance, the methods of reconstitution of botulinum toxin, the angle of injection plane, whether or not medial rectus muscle grasping with forceps, or instructions of patients’ care).

Response

After eyelid speculum placement, patients were instructed to look to the temporal side to expose the nasal conjunctiva. We grasped the nasal conjunctiva with forceps, and the needle was aimed inferonasally at a tangent to the globe with the bevel facing upward. Botulinum toxin was slowly injected for 20 - 30 seconds using a 30-gauge needle into the nasal conjunctiva at approximately 8 - 10 mm from the limbus closely targeted at the belly of the medial rectus muscle. We added this in the methods section. (Lines 93-100)

3. I have two questions about the data present in the table 

1) First, the PD value in the esodeviation at 6 months after injection had a much small standard deviation in the failure group (± 1.6), an explanation is needed. 

Response

This owes to the definition of success and failure. Treatment was considered successful if the ocular alignment was orthotropia or esodeviation of 10 PD or less during distant fixation. Patients with an esodeviation of > 10 PD during distant fixation were classified as the failure group. 

In our study, most of the patients in the failure group had a narrow range of esodeviation at distance from 12 PD to 16PD. On the other hand, the range of distant ocular deviation was between 6PD exodeviation to 10PD esodeviation. 

2) Second, the success group seemed to have a greater change in PD at 2 weeks from baseline compared to the failure group. The dose of botulinum toxin standardized according to the deviation angle (PD/Unit) might be different between success and failure group. A statistics analysis showing no difference between the groups is needed to exclude this possibility. Indeed, a significant post-operative variable as a predictor of post-operative success may be of little clinical significance especially when the variables seemed to be highly correlated with the grouping criteria in a retrospective study design.

Response

Thank you for your pertinent review. We performed a thorough and extensive review regarding our data, and found that an initial postoperative esodeviation of 18 PD or less at one month after exotropia surgery was a significant factor of success with botulinum toxin injection. No other preoperative factor including the dose of botulinum toxin, interval between surgery and toxin injection, or even the baseline angle of esodeviation before injection was associated with treatment success. We added this in the results and Table 2.

4. The results of multivariate logistic analysis could be represented in different methods but should not be partially revealed by a few sentences.

Response

We agree with the reviewer that results of multivariate logistic analysis could be represented in a table. But as most of the variables, including the factors in Table 2, were not significant by univariate analysis, we considered it to be sufficient to describe the results as it is. 

As per the reviewer’s suggestion, we performed a thorough and extensive review regarding our data, and found that the initial postoperative esodeviation at one month after exotropia surgery was also a significant factor of success with botulinum toxin injection. This was also consistent by multivariate analysis, as the initial postoperative esodeviation of 18PD or less at one month after exotropia surgery and immediate success at 2 weeks after botulinum injection were significant factors of success after six months. We added this in the results and discussion. 

We appreciate your precious time and effort for reviewing our manuscript. 

 

Reviewer #3: The authors describe the efficacy and safety of botulinum toxin injection without using EMG for consecutive ET clearly. In addition, the manuscript identifies the predictive factors of success after botulinum toxin injection. Minor points to consider in subsequent versions:

1. The manuscript highlights that immediate success at 2 weeks after botulinum injection was the only significant factor of success. Would you consider to emphasize the result in the Abstract the difference from other similar published literature?

Response

We performed a thorough and extensive review regarding our data, and found that an initial postoperative esodeviation of 18 PD or less at one month after exotropia surgery was also a significant factor of success with botulinum toxin injection. We added this in the abstract and results. 

2. Page 7: The predictive factors were well determined, including previous MR resection. However, the surgical methods for XT was not discussed completely in the study. In current surgical approaches, including unilateral LR recession and MR resection, bilateral LR recession and involving three or four horizontal rectus, which one might get most benefit from botulinum toxin injection? Please discuss the influence of previous surgical methods for XT.

Response

Thank you for your suggestion. We added the type of surgery in Table 1. However, the type of surgery, number of operated horizontal muscles, and the number of operations did not prove to be a predictive factor of success after botulinum injection. (Table 2)

3. Page 6: Seventy-one percent of patients received only one injection with good outcome in your study. Please analyze the risks for repeated injections.

Response

The risk of repeated injection was identical to the risk of failure at six months after injection, because reinjection of botulinum toxin was considered at 6 months after the initial injection (failure group) only if the angle of esodeviation was > 10 PD during distant fixation. However, some patients decided not to receive reinjection and simply maintained prism glasses. Therefore, we determined the failure group as patients who required reinjection (and not the patients who actually performed reinjection). 

4. Page 5: Please describe the procedure of botulinum injection more completely. Do you grip the insertion of MR muscle while injection? Some surgeons hold the needle in place for 5-10 seconds after injection. Do you do the same protocol or not? How do you adjust the distance from limbus (line 86: 8-10 mm mentioned) in children and adults (range of age: 7-66 years in Results)? Is the location of injection the same or different?

Response

The type of botulinum toxin was a liquid premixed injectable toxin without the need for reconstitution. After eyelid speculum placement, patients were instructed to look to the temporal side to expose the nasal conjunctiva. We grasped the nasal conjunctiva with forceps, and the needle was aimed inferonasally at a tangent to the globe with the bevel facing upward. Botulinum toxin was slowly injected for 20 - 30 seconds using a 30-gauge needle into the nasal conjunctiva at approximately 8 - 10 mm from the limbus closely targeted at the belly of the medial rectus muscle. This distance is not quite different between adults and children ≥ 7 years of age.

5. Page 6: The range of interval form previous strabismus surgery to the time of botulinum toxin injection was large (line 113: 4-151 months). What is your consideration in early and later intervention? Please describe in the section of Discussion.

Response

The optimal time and indications for botulinum toxin injection in consecutive esotropia remain unclear.[10] Considering the natural exodrift after exotropia surgery, it is likely that small angles of esotropia may resolve spontaneously over time.[3,4] Park et al. have shown that more than 50% of the total amount of exodrift was observed within the first postoperative year, but no subsequent changes were observed after 3 years.[21] Couser et al. reported that children treated within 12 months of consecutive esotropia had an excellent outcome; however, the patient with the longest delay of 53 months between the onset of the consecutive esotropia and treatment also had an excellent outcome after 2 injections.[10] In our study, the mean interval from strabismus surgery to the time of botulinum toxin injection was 47.3 ± 39.0 months (range, 4 - 151 months), and the patient with the longest delay of 151 months between the time of operation and botulinum treatment also had a successful outcome after two injections. Therefore, botulinum toxin treatment should be actively considered if significant consecutive esotropia persists after 3 years. 

6.Line 308: Please cite the newest reference “ Botulinum toxin for the treatment of strabismus. Cochrane Database Syst Rev. 2017 Mar 2;3:CD006499. doi: 10.1002/14651858. CD006499.pub4.” instead of the 2012 old one.

Response

Yes, we changed the mentioned reference to an updated version. 

We appreciate your precious time and effort for reviewing our manuscript. 

Reviewer #4: This is an interesting research and written properly. However, there are some 

questions need to be answered:

1. How was the dosage determined prior to injection? In line 88, determined by the angle of esodeviation, does it mean how many IU for how many prism diopter?

Response

The total dosage of botulinum toxin ranged from 6 - 12 IU, which was determined empirically by the angle of esodeviation ranging from 6 to 45 prism diopters (PD). (Lines 103-104) As frequent undercorrection was noted, we progressively increased the amount of botulinum toxin injection throughout the study period. Thus, the average dose of botulinum toxin injection in our study was slightly larger than that in previous studies using electromyographic guidance. (Lines 280-283)

2. Did you inject unilateral or bilateral medial rectus muscles?

Response

We injected bilaterally in both medial rectus muscles. (Lines 92)

3. In line 87, 6-12 IU refers to does per injection or total injection per case? Same question goes to line 117

Response

This refers to the total dosage of botulinum toxin injected into both MR muscles per patient. 

4. Finally, do you compare these cases to the natural course? (consecutive ET nature course)

Response

Considering the natural exodrift after exotropia surgery, it is likely that small angles of esotropia may resolve spontaneously over time.[3,4] Park et al. have shown that more than 50% of the total amount of exodrift was observed within the first postoperative year significantly, but no subsequent changes were observed after 3 years.[21] Couser et al. reported that children treated within 12 months of consecutive esotropia had an excellent outcome, however, the patient with the longest delay of 53 months between the onset of the consecutive esotropia and treatment also had an excellent outcome after 2 injections.[10] In our study, the mean interval from strabismus surgery to the time of botulinum toxin injection was 47.3 ± 39.0 months (range, 4 - 151 months) and the patient with the longest delay of 151 months between the time of operation and botulinum treatment also had a successful outcome after two injections. Therefore, botulinum toxin treatment should be actively considered if significant consecutive esotropia persists after 3 years. 

We appreciate your precious time and effort for reviewing our manuscript. 

Reviewer #5: Materials and Methods

1. No information about conservative treatment (watching, patching or prism glasses) before injection. When is the optimal time of botulinum toxin injection for treating a case with consecutive esotropia?

Response

The optimal time and indications for botulinum toxin injection in consecutive esotropia is not clear.[10] Considering the natural exodrift after exotropia surgery, it is likely that small angles of esotropia may resolve spontaneously over time.[3,4] Park et al. have shown that more than 50% of the total amount of exodrift was observed within the first postoperative year significantly, but no subsequent changes were observed after 3 years.[21] Couser et al. reported that children treated within 12 months of consecutive esotropia had an excellent outcome, however, the patient with the longest delay of 53 months between the onset of the consecutive esotropia and treatment also had an excellent outcome after 2 injections.[10] In our study, the mean interval from strabismus surgery to the time of botulinum toxin injection was 47.3 ± 39.0 months (range, 4 - 151 months) and the patient with the longest delay of 151 months between the time of operation and botulinum treatment also had a successful outcome after two injections. Therefore, botulinum toxin treatment should be actively considered if significant consecutive esotropia persists after 3 years. 

2. Line 86 vs Line 116: dosage of botulinum toxin ranged from 6-12 IU vs 3-12 Which one is correct?

Response

I am sorry for the confusion. A dosage of 3 IU was injected into each medial rectus muscles and a total of 6IU for both muscles is correct. We changed the manuscript accordingly. 

3. Line 87: the angle of esodeviation ranging from 6 to 45 PD � So you included the cases whose esodeviation angle before injection is less than 10 PD ?

Response

Yes, this is true for three patients with 6~8PD of esotropia whose angle of esodeviation no longer decreased after wearing prism glasses for a significant time. In these patients, the intervals between the time of surgery and botulinum treatment were 10, 30, and 143 months after surgery. The final results were orthotropia, 6PD esophoria and 6 PD exophoria after injection. All patients were classified as success. However, only the patient with a residual 6PD esophoria had an initial esotropia of 20PD at 1 month after exotropia surgery, while the other two patients had an initial esotropia of 6PD and 14PD esotropia. This also reflects the importance of the amount of initial esotropia within 1 month after surgery as a predictor of success after botulinum treatment. 

Result

1. Line 110: The mean interval from strabismus surgery to the time of botulinum toxin injection was 47.3±39 moths (range, 4-151months). Most patients with consecutive esotropia resolves spontaneously in one year. How many cases with consecutive esotropia have botulinum toxin injection within one year (or 6 months) after strabismus surgery?

Response

Ten patients received injection within one year after strabismus surgery. These patients had a relatively large angle of esodeviation, as large as up to 44PD, which is less likely to spontaneously resolve over time. (Lines 242-245) 

2. Please give more details of the dosage of botulinum toxin, the mean interval from strabismus surgery to the time of first injection, and the amount of esodeviation over time.

Response

The total dosage of botulinum toxin ranged from 6 - 12 IU, which was determined empirically by the angle of esodeviation ranging from 6 to 45 prism diopters (PD). (Lines 103-104) As frequent undercorrection was noted, we progressively increased the amount of botulinum toxin injection throughout the study period. Thus, the average dose of botulinum toxin injection in our study was slightly larger than that in previous studies using electromyographic guidance. (Lines 280-283)

The mean interval from strabismus surgery to the time of botulinum toxin injection was 47.3 ± 39.0 months (range, 4 - 151 months). The mean rate of change in the angle or esodeviation after 6 months to the last follow-up examination was -0.1 ± 2.7 PD/year (range, -8~12). (Lines 153-155)

Discussion

1. You did not discuss how to exclude the possibility of spontaneous recover so that we can evaluate the efficacy of botulinum toxin properly.

Response

Considering the natural exodrift after exotropia surgery, it is likely that small angles of esotropia may resolve spontaneously over time.[3,4] Park et al. have shown that more than 50% of the total amount of exodrift was observed within the first postoperative year significantly, but no subsequent changes were observed after 3 years.[21] Couser et al. reported that children treated within 12 months of consecutive esotropia had an excellent outcome, however, the patient with the longest delay of 53 months between the onset of the consecutive esotropia and treatment also had an excellent outcome after 2 injections.[10] In our study, the mean interval from strabismus surgery to the time of botulinum toxin injection was 47.3 ± 39.0 months (range, 4 - 151 months) and the patient with the longest delay of 151 months between the time of operation and botulinum treatment also had a successful outcome after two injections. Therefore, botulinum toxin treatment should be actively considered if significant consecutive esotropia persists after 3 years. Ten patients received botulinum treatment within one year after strabismus surgery. These patients had a relatively large angle of esodeviation up to 44PD which is less likely to spontaneously resolve over time.

2. More details about the dosage of botulinum toxin still can inspire the readers even though the dose effect cannot be standardized because of various types.

Response

The total dosage of botulinum toxin ranged from 6 - 12 IU, which was determined empirically by the angle of esodeviation ranging from 6 to 45 prism diopters (PD). (Lines 103-104) As frequent undercorrection was noted, we progressively increased the amount of botulinum toxin injection throughout the study period. Thus, the average dose of botulinum toxin injection in our study was slightly larger than that in previous studies using electromyographic guidance. (Lines 277-282) Recently, we generally inject 12 IU in both medial rectus muscles (6 IU each) for patients with an esodeviation of more than 10PD. 

3. Overcorrection with an exotropia of ≥ 10PD at distance was found in 11 patients at two weeks after treatment but resolved within 3 months. This means nearly all overcorrected cases at 2 weeks after treatment are in failure group in the end. What do you think about this? Why dose botulinum toxin only cause permanent impact in some cases?

Response

There must have been some misunderstanding. Patients with an esodeviation of > 10 PD during distant fixation were classified as the failure group at 6 months after injection. All patients who were overcorrected after botulinum treatment (exotropia ≥ 10 PD at 2 weeks) showed a successful outcome at 6 months. Only 1 patient showed failure at the last follow-up examination. 

We appreciate your precious time and effort for reviewing our manuscript. 

 

Reviewer #6: The concept behind the paper is interesting. That is helping ophthalmologists to choose the ideal strategy to deal with the troublesome postoperative consecutive strabismus.

1. There are lots of grammar mistakes in the article. And several repeated sentences noted (Exp. Line 67&104). Please do the grammar check.

Response

Thank you for your careful review. We sent the manuscript for a professional English editing service. 

2. The authors need to clarify in the ethics approval the IRB number.

Response

The IRB number is as follows: IRB NoB-1905/541-101. We added this in the manuscript. 

3. The authors need to explain the rational why they performed “subconjunctival injection” rather than “subtenon injection”.

Response

Botulinum was partly injected in the subtenon space as we targeted the drug to disperse as near as possible to the medial rectus muscle. However, as we could not be sure of the exact position of the drug after injection without the use of electromyographic guidance, we used the term “subconjunctival”. 

4. Is there any association between the pre-op deviation/surgical procedure and the injection outcome?

Response

The rate of patients with an initial postoperative esodeviation of 18PD or less at one month after exotropia surgery was significantly larger in the success group compared to the failure group (70.3% vs 33.3%, P = 0.014). By multivariate analysis, an initial postoperative esodeviation of 18PD or less at one month after exotropia surgery (P = 0.012) and immediate success at 2 weeks after botulinum injection were significant factors of success after six months (P = 0.015). We added this in the manuscript. 

5. The short-term complication rate in this study is very low (7.7%). Ptosis/vertical deviation/overcorrection presented much frequently in literature (21-33%). Please share the pearls of reducing complications?

Response

We described the procedure in detail as follows: 

“The type of botulinum toxin was a liquid premixed injectable toxin without the need for reconstitution. After eyelid speculum placement, patients were instructed to look to the temporal side to expose the nasal conjunctiva. We grasped the nasal conjunctiva with forceps, and the needle was aimed inferonasally at a tangent to the globe with the bevel facing upward. Botulinum toxin was slowly injected for 20 - 30 seconds using a 30-gauge needle into the nasal conjunctiva at approximately 8 - 10 mm from the limbus closely targeted at the belly of the medial rectus muscle. The same procedure was performed in both medial rectus muscles. The total dosage of botulinum toxin injection in both medial rectus muscles per patient ranged from 6 - 12 IU, which was determined empirically based on their esodeviation, which ranged from 6 to 45 prism diopters (PD). Patients were instructed not to rub their eyes and to remain upright for at least one hour.”

6. The success rate in the study is excellent. However, the 6-month-f/u time is too early to judge the long-term success. I would like to know the outcome after 12-month f/u.

Response

We also did a subgroup analysis of a 12-month FU regarding 35 patients. However, as the number of patients were small in the success group and failure group, significant factors were not revealed in this case. A longer study period consisting of a larger number of patients would be necessary to determine the long-term efficacy of this procedure. 

We appreciate your precious time and effort for reviewing our manuscript. 

 

Reviewer #7: The authors conducted a retrospective study on 52 subjects with consecutive esotropia who underwent botulinum toxin injection in the medial rectus muscles without the use of electromyographic guidance. At 6-months, 71.2% showed successful alignment, which is comparable to unilateral medial rectus muscle recession surgery. They also found immediate success at 2 weeks after botulinum injection is predictive of success after 6 months.

Comments

1. Did the authors investigate the type of surgery (bilateral lateral rectus muscle recession or unilateral lateral rectus muscle recession + medial rectus muscle resection/plication) before botulinum injection affect the success?

Response

Yes, we investigated the type of surgery and added the demographics in Table 1. However, there was no difference in the type of surgery between the success group and failure group.

2. Why did the authors use such Botulinum doses, which is slightly larger than previous studies? Does the injection dose follow the same protocol?

The dosage of injection was chosen empirically which was not consistent throughout the study. As frequent undercorrection was noted, we progressively increased the amount of botulinum toxin injection throughout the study period. Even in patients who underwent previous unilateral lateral rectus recession and medial rectus resection, we injected botulinum toxin in both medial rectus muscles. This was based on the fact that frequent undercorrection was encountered even with a relatively large dose (7 IU) of botulinum toxin injection into only one medial rectus muscle in the earlier periods before this study. Recently, we generally inject 12 IU in to both medial rectus muscles for patients with an esodeviation of more than 10 PD.

We appreciate your precious time and effort for reviewing our manuscript.

---

## [Decision Letter · Decision Letter 1]

3 Aug 2020

PONE-D-20-07281R1

­­­­­­­­­­­­­­­­­­­Botulinum toxin injection without electromyographic guidance in consecutive esotropia

PLOS ONE

Dear Dr. Hwang,

Thank you for submitting your manuscript to PLOS ONE. After careful consideration, we feel that it has merit but does not fully meet PLOS ONE’s publication criteria as it currently stands. Therefore, we invite you to submit a revised version of the manuscript that addresses the points raised during the review process.

We look forward to receiving your revised manuscript.

Kind regards,

I-Jong Wang

Academic Editor

PLOS ONE

Reviewers' comments:

Reviewer's Responses to Questions

**Comments to the Author**

1. If the authors have adequately addressed your comments raised in a previous round of review and you feel that this manuscript is now acceptable for publication, you may indicate that here to bypass the “Comments to the Author” section, enter your conflict of interest statement in the “Confidential to Editor” section, and submit your "Accept" recommendation.

Reviewer #1: All comments have been addressed

Reviewer #2: All comments have been addressed

Reviewer #4: All comments have been addressed

Reviewer #5: All comments have been addressed

Reviewer #6: All comments have been addressed

2. Is the manuscript technically sound, and do the data support the conclusions?

Reviewer #1: Partly

Reviewer #2: Yes

Reviewer #4: Yes

Reviewer #5: Yes

Reviewer #6: Yes

3. Has the statistical analysis been performed appropriately and rigorously? 

Reviewer #1: Yes

Reviewer #2: Yes

Reviewer #4: Yes

Reviewer #5: Yes

Reviewer #6: Yes

4. Have the authors made all data underlying the findings in their manuscript fully available?

Reviewer #1: Yes

Reviewer #2: Yes

Reviewer #4: Yes

Reviewer #5: Yes

Reviewer #6: Yes

5. Is the manuscript presented in an intelligible fashion and written in standard English?

Reviewer #1: Yes

Reviewer #2: Yes

Reviewer #4: Yes

Reviewer #5: Yes

Reviewer #6: Yes

6. Review Comments to the Author

Reviewer #1: Dear Authors,

Thanks for your responses and explanations in detail.

Botulinum injection without EMG guide indeed needs greater care of the localization of the injection site very precisely. Thanks for sharing this method and your outcomes in treating consecutive esotropia with us. There are several points need to be explained more clearly.

1. As the authors mentioned, the dosage of the Botulinum injection differed throughout the whole study. Therefore, it is difficult to compare them all on the same basis. Please persuade us that they are comparable in the discussion.

2. How about classifying them into subgroups of different doses of Botulinum to see if there are any differences in the success/failure rate and to clarify the optimal dosage.

3. The differences of the dosages of Botulinum and treatment outcomes between previous EMG-guided injection studies and the authors' study should be mentioned.

Reviewer #2: The authors had adequately addressed all my comments and revised their manuscript. Only two trivial questions remained.

1.In table 2, the variables "Immediate success at 2 weeks after injection " is duplicated.

2.The revised manuscripts showed that initial postoperative esodeviation of less than 18 PD at one month after exotropia remained a significant predictive factor of success. I am interested in how the authors determine this cu-toff value of PD. (for example, via ROC curve analysis ?)

Reviewer #4: The author has answered most of the questions proposed, however, in table 2, the last line duplicates the 3rd line and should be revised.

Reviewer #5: It is great that those 3 patients with 6-8 PD of esotropia before injection had improved after Botulinum toxin injection. However, it is controversial to include these 3 cases who meet the definition of the success group at the beginning.

Reviewer #6: Thank you for the revision.

Please clarify the Esodeviation at near before injection in the abstract.

7. PLOS authors have the option to publish the peer review history of their article (what does this mean?). If published, this will include your full peer review and any attached files.

Reviewer #1: No

Reviewer #2: No

Reviewer #4: No

Reviewer #5: No

Reviewer #6: No

---

## [Author Response · Author response to Decision Letter 1]

10 Sep 2020

September 8, 2020

I-Jong Wang

Academic Editor

PLOS ONE

Ref: PONE-D-20-07281R2

¬¬¬¬¬¬¬¬¬¬¬¬¬¬¬¬¬¬¬Botulinum toxin injection without electromyographic guidance in consecutive esotropia

Dear Dr. I-Jong Wang:

I thank the editors and reviewers for taking their time to review our manuscript entitled "¬¬¬¬¬¬¬¬¬¬¬¬¬¬¬¬¬¬¬Botulinum toxin injection without electromyographic guidance in consecutive esotropia”. A response to the reviewers’ comments/requests in accordance with the Editorial Board Member’s recommendation is included below. Each of the coauthors has seen and agreed with the changes made to this revision. I hope the revised manuscript will better meet the publication requirement of PLOS ONE.

Sincerely,

Jeong-Min Hwang, M.D., PhD

Professor

Department of Ophthalmology, 

Seoul National University College of Medicine

Seoul National University Bundang Hospital

1. Please provide additional details regarding participant consent. In the Methods section, please ensure that you have specified (1) whether consent was informed and (2) what type you obtained (for instance, written or verbal). If your study included minors, state whether you obtained consent from parents or guardians. If the need for consent was waived by the ethics committee, please include this information.

We mentioned the contents in the method section as follows: 

Written consent was obtained from the patient or parent and/or legal guardian for botulinum toxin injection. This study was conducted in compliance with the Declarations of Helsinki and was approved by the Institutional Review Board of Seoul National University Bundang Hospital (IRB number: B-1905/541-101).

---

## [Decision Letter · Decision Letter 2]

19 Oct 2020

­­­­­­­­­­­­­­­­­­­Botulinum toxin injection without electromyographic guidance in consecutive esotropia

PONE-D-20-07281R2

Dear Dr. Jeong-Min Hwang,

We’re pleased to inform you that your manuscript has been judged scientifically suitable for publication and will be formally accepted for publication once it meets all outstanding technical requirements.

Kind regards,

I-Jong Wang

Academic Editor

PLOS ONE

Additional Editor Comments (optional):

Reviewers' comments:

Reviewer's Responses to Questions

**Comments to the Author**

1. If the authors have adequately addressed your comments raised in a previous round of review and you feel that this manuscript is now acceptable for publication, you may indicate that here to bypass the “Comments to the Author” section, enter your conflict of interest statement in the “Confidential to Editor” section, and submit your "Accept" recommendation.

Reviewer #1: All comments have been addressed

Reviewer #2: All comments have been addressed

Reviewer #4: (No Response)

Reviewer #5: (No Response)

Reviewer #6: All comments have been addressed

2. Is the manuscript technically sound, and do the data support the conclusions?

Reviewer #1: Yes

Reviewer #2: Yes

Reviewer #4: Yes

Reviewer #5: Yes

Reviewer #6: Yes

3. Has the statistical analysis been performed appropriately and rigorously? 

Reviewer #1: Yes

Reviewer #2: Yes

Reviewer #4: Yes

Reviewer #5: Yes

Reviewer #6: Yes

4. Have the authors made all data underlying the findings in their manuscript fully available?

Reviewer #1: Yes

Reviewer #2: Yes

Reviewer #4: Yes

Reviewer #5: Yes

Reviewer #6: Yes

5. Is the manuscript presented in an intelligible fashion and written in standard English?

Reviewer #1: Yes

Reviewer #2: Yes

Reviewer #4: Yes

Reviewer #5: Yes

Reviewer #6: Yes

6. Review Comments to the Author

Reviewer #1: (No Response)

Reviewer #2: Thanks for your responses to the rest of my questions. I think the manuscript is ready for publication.

Reviewer #4: (No Response)

Reviewer #5: (No Response)

Reviewer #6: The authors had adequately addressed all my comments and revised their manuscript.

I have no further comments.

7. PLOS authors have the option to publish the peer review history of their article (what does this mean?). If published, this will include your full peer review and any attached files.

Reviewer #1: No

Reviewer #2: No

Reviewer #4: No

Reviewer #5: No

Reviewer #6: No

---

## [Editor Report · Acceptance letter]

4 Nov 2020

PONE-D-20-07281R2 

Botulinum toxin injection without electromyographic guidance in consecutive esotropia 

Dear Dr. Hwang:

I'm pleased to inform you that your manuscript has been deemed suitable for publication in PLOS ONE. Congratulations! Your manuscript is now with our production department. 

Kind regards, 

on behalf of

Dr. I-Jong Wang 

Academic Editor

PLOS ONE